# Feasibility, Acceptability, and Preliminary Validity of Self-Report Dietary Assessment in Adults with Multiple Sclerosis: Comparison with Doubly Labeled Water Measured Total Energy Expenditure

**DOI:** 10.3390/nu13041198

**Published:** 2021-04-05

**Authors:** Stephanie L. Silveira, Brenda Jeng, Barbara A. Gower, Robert W. Motl

**Affiliations:** 1Department of Physical Therapy, University of Alabama at Birmingham, 3810 Ridgeway Drive, Birmingham, AL 35209, USA; bjeng@uab.edu (B.J.); robmotl@uab.edu (R.W.M.); 2Department of Kinesiology, Health Promotion, and Recreation, University of North Texas, 1155 Union Circle #310769, Denton, TX 76203, USA; 3Department of Nutrition Sciences, University of Alabama at Birmingham, 1675 University Blvd, Birmingham, AL 35233, USA; bgower@uab.edu

**Keywords:** multiple sclerosis, diet, nutrition, doubly labeled water, energy intake, energy expenditure

## Abstract

Background: Diet is a modifiable behavior of interest in multiple sclerosis (MS); however, measures of diet in persons with MS have not been vetted for feasibility, acceptability, and validity. Methods: This cross-sectional study examined the Automated Self-Administered 24-H (ASA24) Dietary Assessment Tool in 30 persons with MS and 15 healthy control (HC) participants. Participants were prompted to complete six ASA24 recalls and undergo a standard doubly labeled water (DLW) protocol. Acceptability of ASA24 was assessed using an online questionnaire. Total energy expenditure (TEE) from DLW was compared with ASA24-reported energy intake for assessing validity. Results: All participants completed four or more ASA24 recalls, indicating feasibility of ASA24. Regarding acceptability, the hardest part of completing the ASA24 was remembering everything eaten the previous day. Pearson correlation coefficients between DLW TEE and ASA24 kcal/day were not significant among HC (r = 0.40; *p* = 0.14) or MS (r = 0.26; *p* = 0.16) participants. The absolute mean error between DLW TEE and ASA24 among HC participants was 694.96 ± 506.25 mean kcal/day and among MS participants was 585.37 ± 529.02 mean kcal/day; this represents a mean difference of 30 and 25%, respectively. Conclusion: This study established the feasibility and acceptability of ASA24 in persons with MS and provides a foundation regarding the need for further validation research examining appropriate outcomes for supporting dietary interventions.

## 1. Introduction

Multiple sclerosis (MS) is chronic neurologic disease, wherein immune mediated degradation of myelin leads to disruptions in communication between the central nervous system and other parts of the body [1]. Common MS symptoms include cognitive impairment, fatigue, and mobility disability and are traditionally treated with pharmacologic therapies [2,3]. Recent evidence suggests that persons with MS are interested in wellness approaches including diet, physical activity, and emotional self-management as second-line therapies for MS symptom and disease management [4,5]. Of those, diet was the number one searched wellness approach among persons with MS and is a modifiable behavior of interest in epidemiological, disease modification, and clinical rehabilitation research [4].

One pressing gap in the literature is the lack of focal research examining diets that may guide appropriate food choices and behavioral approaches in MS. Indeed, interventions focused on diet in MS may target inflammation, protect against neurodegeneration, and promote nervous system repair [6]. Calorie restriction, as an example, has been posited as beneficial in MS by increasing endogenous corticosteroid production, decreasing inflammatory cytokines, and increasing neurotrophic factors [7]. There is a major gap in the literature that precludes the application of calorie restriction, or any other dietary interventions, in persons with MS, namely measurement.

To our knowledge, there are no published studies validating dietary assessment, specifically energy intake, in persons with MS. The few studies examining diet in persons with MS utilize self-report measures including food frequency questionnaires, food diaries, diet screening questionnaires, and brief dietary assessment instruments [8,9,10,11]. Such self-report measures have been identified as inappropriate measures of energy intake in persons with cognitive impairment (e.g., elderly individuals) and may not be valid in the MS population based on the presence of memory and information processing impairments [12]. There is a pressing need to identify appropriate tools for measuring diet in MS that can be applied for assessing compliance with dietary interventions.

The current study examined the feasibility, acceptability, and validity of 24 h dietary recall protocol self-administered online using the Automated Self-Administered 24-H (ASA24) Dietary Assessment Tool. We located two studies that have applied the ASA24 as a measure of energy intake in persons with MS (one cross-sectional and one intervention study) [13,14]; however, the feasibility, acceptability, and validity of this measure has not been established in persons with MS. Validity, in particular, was examined through comparison of self-reported energy intake on the ASA 24 with the gold standard measure of total energy expenditure, doubly labeled water (DLW). We hypothesized that ASA24 will be a feasible and acceptable method of measuring energy intake in persons with MS. We further hypothesized imprecise measurement of energy intake when comparing self-report energy intake from ASA24 with total energy expenditure from DLW consistent with the general population literature [12].

## 2. Materials and Methods

### 2.1. Participants

Participants were recruited through convenience sampling methods. Participants with MS who previously completed studies in our research laboratory were mailed flyers advertising the study. Interested participants were prompted to e-mail or call the research team for a telephone screening. Telephone screening included an overview of the study procedures and questions assessing the inclusion criteria: (a) diagnosis of MS; (b) relapse-free for the past 30 days; (c) willingness to complete two study visits including questionnaires, 6 ASA24 dietary recalls, and DLW protocol; (d) age between 18 and 55 years; (e) access to Internet and e-mail for ASA24 completion. Exclusion criteria included self-report weight loss of 10 or more pounds over the past 3 months to ensure weight stability. We recruited healthy controls (HC) without MS who matched a participant with MS regarding biological sex and age within 5 years and further met inclusion criteria c, d, and e as well as exclusion criteria. HCs without MS were recruited via an advertisement on a University Research website. Our a priori recruitment goal included 30 participants with MS and 15 healthy controls (i.e., 2:1 sampling), and this aligned with previous feasibility studies [15], rather than a power analysis, and the limitations of subject associated with the pilot funding mechanism.

### 2.2. Procedure

Upon completion of telephone screening, eligible participants were scheduled for the first study visit. Participants completed informed consent and then a standard DLW protocol. Participants completed a battery of questionnaires and the first ASA24 recall on an iPad. After completion of the first ASA24 recall, participants completed an acceptability questionnaire regarding experiences completing the first ASA24 recall. The baseline assessment was not complete until participants provided a final 4 h urine sample collection. Participants were prompted via e-mail or text on 2 random days following the baseline assessment to complete 2 additional ASA24 recalls (3 ASA24 diet recalls within one week of baseline). Participants were asked to return 14 days following the baseline to provide a urine sample for DLW protocol and complete questionnaires that included an additional acceptability questionnaire regarding experiences completing the ASA24 recalls and a 4th ASA24 recall. Participants were then prompted to complete 2 additional ASA24 diet recalls the week following the 14-day assessment (3 ASA24 diet recalls within one week of 14-day assessment). Participants were provided a USD 75 gift card for completing each assessment visit for a total potential sum of USD 150.

### 2.3. Measures

#### 2.3.1. Biologically Measured Total Energy Expenditure

The doubly labeled water (DLW) method was used to assess total energy expenditure (TEE) based on established procedures [16]. Participants self-reported body weight during telephone screening and body weight was verified at baseline assessment visit for isotope calculation. At the baseline visit, a baseline urine sample was collected prior to oral administration of a mixed dose of DLW and 50 mL of tap water for rinsing the DLW administration cup. The solutions used were Cortecnet Oxygen-18 (H_2_^18^O) Isotope Enrichment ≥10% and Deuterium (D^2^O) 99.8% atom D. Further, the dose administered was calculated: ^18^O:D_2_O is 1 g:0.08 g by weight, wherein participants were dosed 1 g total solution per kg = 0.926 g/kg of water with 10% ^18^O atoms and 0.074 g/kg of water with 99.8% ^2^H atoms [17]. Participants were instructed to void their bladder to the best of their ability, in line with standard DLW protocol, before the baseline urine samples. Second and third urine samples were obtained following 3 h and 4 h equilibration, respectively. The final urine sample was obtained at 14 days; however, two participants had scheduling conflicts that required samples on alternate days, one on Day 12 and one on Day 13. Date and time of all samples were collected to ensure appropriate calculations were applied. Samples were analyzed in duplicate for H_2_^18^O and ^2^H_2_O enrichments by isotope ratio mass spectrometry (IRMS) on a Thermo Scientific Delta V Advantage IRMS with Gas Bench. Turnover rates and zero-time extrapolated dilution spaces of H^2^_18_O and ^2^H_2_O were calculated from the slope and intercept of the semi-logarithmic plot of isotope enrichment in urine, versus time after dosing, using the Coward Equations [18]. CO_2_ production rates and TEE were calculated based on the recent updated equations of Speakman et al. [19]. TEE is expressed in this report as kcal/day. Equations for each are included below:Dilution space = F_1_N_1_ = F_2_N_2_; N_2_ = F_1_N_1_/F_2_(1)
where N_1_ is the pool size of the dose in moles, and N_2_ is the pool size of the distribution space in moles. F_1_ is enrichment of the dose, and F_2_ is the enrichment of the distribution space.
rCO_2_ = 0.4554 * N [1.007 * k_o_)] * 22.26(2)
TEE (MJ/d) = rCO_2_ * (1.106 + (3.94/RQ) * (4.184/10^3^)(3)

#### 2.3.2. Self-Reported Energy Intake

The Automated Self-Administered 24-H (ASA24) Dietary Assessment Tool (version 2018), developed by the National Cancer Institute, was used to assess energy intake [20]. The ASA24 is a multi-pass, self-administered 24 h recall method that is administered through a secure online website. A comprehensive overview regarding the foundational validation studies of the ASA24 have been previously published [21]. Briefly, participants are guided through a series of steps aimed to capture all food and beverage intake during the previous 24 h period. Participants completed their first ASA24 during the baseline assessment where a trained researcher was available to answer questions. All participants were then prompted via e-mail or text to complete 2 additional ASA24 diet recalls within one week of their baseline assessment on random, non-consecutive days. A similar protocol was applied at the 14-day assessment in which participants completed their 4th ASA24 diet recall during the appointment and then prompted to complete 2 additional ASA24 diet recalls within one week of the 14-day assessment on 2 random, non-consecutive days. The ASA24 prompts were scheduled to align with best practices in diet recall methodology, specifically 3 diet recalls during a one-week period that are non-consecutive and include 2 weekdays and one weekend day [22]. All valid days were combined for each participant, and this yielded a mean energy intake value, irrespective of weekday or weekend days. Based on previous research, any days in which the mean reported energy intake was below 500 kcals were considered erroneous and were excluded [22,23].

#### 2.3.3. Acceptability Questionnaire

Participants completed a brief, researcher-developed questionnaire to assess the acceptability of ASA24 dietary recalls. Participants completed one set of questions immediately following their first ASA24 at baseline that included the following questions: have you ever used the ASA24 online system? (multiple choice: MC); how difficult was it to remember everything you ate yesterday? (MC); how did you feel using the ASA24 online system? (MC); what do you think would have helped make it easier to complete the ASA24 dietary recall? (open ended: OE). An additional measure of acceptability was administered at the 14-day assessment prior to completing their 4th ASA24 that included the following questions: how difficult was it to remember everything you ate on previous days? (MC); how easy was it to access the ASA24 online system? (MC); how did you feel using the ASA24 online system? (MC); what was the HARDEST part of completing the ASA24 online dietary recall? (MC); what was the EASIEST part of completing the ASA24 online dietary recall? (MC); how helpful did you find the e-mail/text reminders? (MC); how did you complete MOST of your ASA24 dietary recalls? (MC); did you need someone to help you complete the dietary recalls? (MC); which method do you think is best to complete a 24 h dietary recall? (MC); what do you think would have helped make it easier to complete the ASA24 dietary recall? (OE)

#### 2.3.4. Demographic and Clinical Characteristics

Participants self-reported biological sex, marital status, age, employment status, race, and level of education. Participants with MS further reported MS clinical course, year of MS diagnosis, and disability status using the Patient Determined Disease Steps [24]. Participants were weighed at each study visit to ensure weight stability.

### 2.4. Data Analyses

Data were analyzed using IBM SPSS Statistics for Windows, version 27 (IBM Corp., Armonk, NY, USA). We examined the differences in demographic characteristics and acceptability questions between HC and MS samples using independent samples *t*-tests and chi-square tests as appropriate. Descriptive statistics, including frequency and percentage, were calculated for the primary research questions regarding feasibility and acceptability of ASA24. ASA24 daily mean energy intake values were averaged per participant across available days of complete data. ASA24 and DLW TEE values were assessed for normality using the Shapiro–Wilk test, and outliers were assessed using box and whiskers plots. Pearson’s correlation coefficient (*r*) analyses were applied for assessing the association between ASA24 and DLW TEE (i.e., preliminary validity). The differences between DLW TEE and average daily energy intake reported in ASA24 were calculated per participant (i.e., absolute error), and a Bland Altman plot with mean absolute error and limits of agreement established the agreement between ASA24 and DLW energy intake values.

## 3. Results

### 3.1. Participants

The participant characteristics are provided in Table 1. The MS group included 30 participants and the HC group included 15 participants. There were no significant differences in demographic characteristics between HC and MS groups.

### 3.2. DLW Protocol Fidelity

All participants completed the DLW protocol. Participants in the HC group did not report any challenges in completing the DLW protocol, whereas five participants in the MS group experienced significant difficulties. Two MS participants were unable to complete the fasted baseline urine sample on the first visit and were rescheduled for another day; both participants were able to complete the protocol on the rescheduled day. Three participants with MS required 30–90 min to produce the fasted baseline urine sample, but completed the full protocol. Following baseline, urine was collected at 3 and 4 h intervals to ensure the research team had at least one post-administration urine sample. Given the challenges in completing urine samples among the first 20 participants, 125 mL of water was administered at the baseline urine collection for participants 21–45 as opposed to the standard 50 mL to rinse the administration cup. DLW TEE results are based on 3 h urine sample values due to availability for all participants. All 45 participant samples were analyzed; however, results for two participants with MS were beyond reasonable values based on this population (>3500 kcal/day and >2SD above sample mean) and were excluded from the validity assessment based on box and whiskers plots. The two participants with invalid DLW assessments did not include any of those who were unable or delayed in providing urine samples.

### 3.3. ASA24 Feasibility

Thirty-four participants completed all six ASA24 recalls, and all participants completed four or more ASA24 recalls. One MS participant and one HC participant had used the ASA24 system before. The overall trends were similar between groups with five HC participants completing 4–5 ASA24 recalls and six MS participants completing 4–5 ASA24 recalls. One participant missed one ASA24 after each assessment. Participants were more likely to skip ASA24 recalls after the 14-day assessment (*n* = 8) as opposed to directly following baseline assessment (*n* = 4).

### 3.4. Acceptability

Acceptability of the ASA24 dietary recall website and protocol was assessed using questionnaires at the baseline and 14-day assessments. Results from each survey are reported in Table 2. All participants reported being very comfortable or comfortable completing the ASA24 at baseline assessment. The hardest part of completing the ASA24 was consistently remembering food you ate, and the easiest part of completing the ASA24 was using the website in both MS and HC participants. There were some notable differences in responses between the MS and HC participants. Five MS participants reported that it was moderately or very difficult to remember everything consumed the day before, whereas no HC participants reported difficulties. Three MS participants reported needing assistance from another person to complete the questionnaire, and two MS participants reported it was not at all easy to access the website. All HC participants reported the online system worked best for them, whereas three participants with MS preferred a phone interview administration, and two participants preferred in-person administration.

Text/e-mail reminders were reported as at least moderately helpful by all participants. Most participants completed the ASA24 on a smartphone, followed by desktop computer. Open-ended responses indicated that participants would have preferred to know in advance the days for completing the recalls, more streamlined entry as opposed to the multi-pass format, and a more robust library of food items and recipes. Suggestions for improving compliance were to have more frequent prompts such as three times a day, ability to log food throughout the day, and suggestions to keep a written food journal.

### 3.5. Validity

We assessed validly of overall mean ASA24 kcal/day through comparison with DLW TEE among the valid cases for both measure (i.e., HC, *n* = 15; MS, *n* = 28). Pearson correlation coefficients between DLW TEE and ASA24 mean kcal/day were not significant among HC participants *r* = 0.40 (*p* = 0.14) or MS participants *r* = 0.26 (*p* = 0.16); however, the correlations were within the typical small-to-moderate magnitude range reported in previous literature [25]. The absolute mean error among HC participants was 694.96 ± 506.25 kcals and among MS participants 585.37 ± 529.02 kcals; this represents a mean percent difference of 30.3 and 24.6%, respectively. The absolute mean error and limits of agreement were then represented and examined using Bland Altman plots for outliers. Among the 28 MS participants, one outlier was identified above the upper limit of agreement (1622 kcal) and no outliers below the lower limit of agreement (−452 kcal) (Figure 1). Among the 15 HC participants, two outliers were identified with one above the upper limit (1687 kcal) and one below the lower limit (−297 kcal). Correlation analyses were conducted to assess for potential bias, and the results did not indicate a significant bias among HC participants *r* = 0.42 (*p* = 0.12) or MS participants *r* = 0.08 (*p* = 0.68) (Figure 2).

## 4. Discussion

The current study established the feasibility, acceptability, and preliminary validity of ASA24 recalls for assessing energy intake in persons with MS. All participants with MS completed the full study protocol, and the percent difference between ASA24 and DLW measures (i.e., 25%) was comparable with validation studies in the general population and smaller than in the control sample [25,26]. Participants provided suggestions for ASA24 recall protocol that should be considered in future applications such as multiple reminders on recall days and options for completion with an interviewer over the phone or face-to-face. Overall, we assert that the ASA24 may be an appropriate tool for assessing energy intake within dietary interventions among persons with MS, particularly when DLW measures are not feasible.

This study established the feasibility and acceptability of ASA24 recalls for dietary assessment in persons with MS. All participants completed four or more of the six ASA24 recalls and rated using the system at baseline as very comfortable or comfortable. Participants with MS appear to have navigated greater challenges completing the ASA24 recalls in the home, wherein five participants reported a preference for an interviewer to administer the recall over the phone or face-to-face rather than the online system. Three participants with MS reported needing some assistance from another person to complete the ASA24 recalls, and the most common challenge was remembering everything consumed the previous day. These results are not surprising given the prevalence and impact of cognitive impairment among persons with MS, particularly memory dysfunction [27,28]. Participants suggested some solutions for improving the ASA24 protocol that included knowing the scheduled recall days in advance with options for continuous entry throughout the day, numerous reminders on recall days to complete the ASA24, and suggestions for participants to keep a food journal during the week of ASA24 recalls. Further research is needed regarding these suggestions given these depart from the standard multi-pass diet recall practices that emphasize the need for unpredictability in recalls to ensure that knowledge of recall days does not influence intake [20]. Such strategies may yield memory recall benefits for persons with cognitive impairments that could outweigh potential bias.

The validity of energy intake measurement using ASA24 in persons with MS was an overarching goal of this study, as we sought to establish it as an appropriate tool for measuring compliance with dietary interventions. DLW-measured TEE was compared with average ASA24 recall-reported energy intake on 4–6 days. Mean absolute differences between ASA24 energy intake and DLW TEE among participants with MS was 24%, and this is aligned with previous validation studies in the general population and was smaller than observed in the control sample [25,26]. Among participants with MS, only one absolute error value was outside the limits of agreement; however, there is still a substantial area that may warrant alternate approaches. Further, DLW TEE and ASA24 energy intake were not correlated in this sample. The sample size was based on previous feasibility research rather than a power analysis as well as the financial constraints of the pilot funding mechanism, and therefore, lack of power may have resulted in non-significant correlation coefficients. An additional important area for further inquiry is fully powered studies that identify factors that contribute to total energy expenditure such as levels of physical activity or cognitive impairment that may account for differences in the magnitude of correlations between energy intake and expenditure.

This report provides details regarding the first application of DLW in participants with MS. All participants completed a baseline fasted urine sample, post-administration 3 or 4 h urine sample, and approximately 14-day urine sample. Participants with MS experienced some challenges in completing the DLW protocol, specifically urine collection in a fasted state, which are likely attributable to bowel and bladder dysfunction among persons with MS. Our research team included an additional, standard water administration 3 h following the DLW administration that improved the overall adherence with the 4 h urine collection; however, the primary challenge among participants with MS was completing the baseline fasted urine sample. One potential avenue for navigating this challenge would be home-based DLW protocols that are currently being investigated (NCT03499509). Such protocols provide participants with the standard DLW protocol instructions and monitor administration via video conferencing software to ensure rigor. Researchers should first assess the validity of home-based versus standard laboratory protocols prior to the application of home-based protocols in participants with MS given more research is warranted to reproducing validity results from this study. An additional option that may be examined in future DLW studies in MS is the use of saliva or blood samples that has potential to overcome challenges associated with urine collection.

This study provides preliminary assessment of DLW protocol and ASA24 recalls as measures of energy intake in persons with MS; however, we note several important limitations. This study included adults with MS 18–55 years old to control for aging-related physiological changes and, therefore, may not generalize among older adults with MS. The first 20 participants were recruited prior to March 2020, whereas the following 25 were recruited from June–August 2020 during the COVID-19 pandemic. Given the restrictions and changes in energy intake during this time period, we assert that mean energy intake values may not be representative of time periods prior to the pandemic. Although all participants in this study completed the DLW protocol, several participants encountered barriers related to bladder dysfunction that could be an indication of dehydration and should be focally examined in future research studies. Participants were sometimes unable to find foods in the ASA24 database and, in such cases, were instructed to report a food of similar composition calorically. The ASA24 uses the gold standard Food and Nutrient Database for Dietary Studies; however, guidance regarding appropriate analogues for foods that may not be available in the database would be useful for future studies. Additionally, the current study assessed the feasibility and acceptability of the ASA24, and the assessment of nutritional status and variation by weekday and weekend days was beyond the scope of the study aims; focal assessment of nutritional status and variation is an area for applying the ASA24 in future research.

## 5. Conclusions

Persons living with MS identify diet as a promising alternative therapy and are interested in diets that may improve symptoms and disease course. The current study ascertained the feasibility and acceptability of ASA24 diet recalls in persons with MS and provides a foundation regarding appropriate outcome tools for supporting dietary interventions. Preliminary validity of ASA24 diet recalls indicates an average of 24% error when compared to DLW TEE that warrants further inquiry in larger samples. Further research is warranted regarding innovative strategies for improving assessment of diet in persons with MS given the prevalence of bladder and cognitive dysfunction that may apply for other neurological conditions such as Parkinson’s, Alzheimer’s, and Huntington’s disease.

## Figures and Tables

**Figure 1 nutrients-13-01198-f001:**
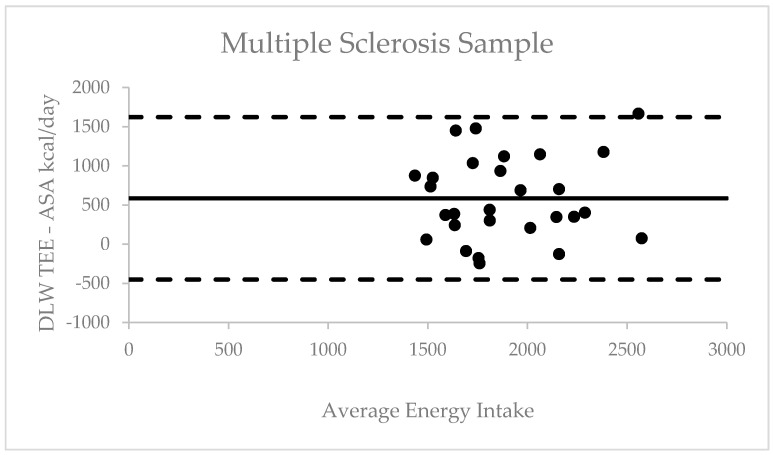
Bland Altman plot of difference between total energy expenditure and self-reported energy intake against average in multiple sclerosis sample.

**Figure 2 nutrients-13-01198-f002:**
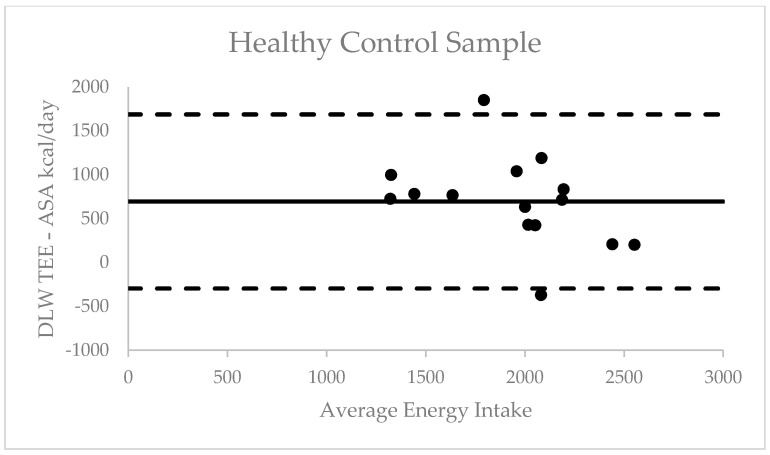
Bland Altman plot of difference between total energy expenditure and self-reported energy intake against average in healthy control sample.

**Table 1 nutrients-13-01198-t001:** Participant demographic and clinical characteristics.

Variable	Healthy Control (HC) Sample (*n* = 15)	Multiple Sclerosis (MS) Sample (*n* = 30)	*p* Value
**Age**, *years ± SD*	34.53 ± 8.77	40.47 ± 9.64	0.05
**Sex**, *n* (%)			0.77
Female	13(86)	25(83)	
Male	2(13)	5(17)	
**Marital Status**, *n* (%)			0.39
Married	5(33)	14(47)	
Single	8(53)	11(37)	
Divorced/Separated	2(13)	4(13)	
Widower	0(0)	1(3)	
**Employment**, *n* (%)			0.22
Yes	13(86)	21(70)	
No	2(13)	9(30)	
**Race**, *n* (%)			0.06
Caucasian	11(73)	13(43)	
African American	4(27)	16(53)	
Other	0(0)	1(3)	
**Education**, *n* (%)			0.50
Less than college degree	6(40)	9(30)	
College degree or more	9(60)	21(70)	
**MS Duration**, *years ± SD*	N/A	10.00 ± 6.24	N/A
**Type MS**, *n* (%)	N/A		N/A
Relapsing Remitting		28(93)	
Progressive		2(7)	
**Patient Determined Disease Steps**, *Median (IQR)*	N/A	0.5(2.0)	N/A

**Table 2 nutrients-13-01198-t002:** Automated Self-Administered 24-H (ASA24) Recall Acceptability Questionnaire results.

Question Response Options	Healthy Control (HC) Sample *n*(%)	Multiple Sclerosis (MS) Sample *n*(%)	Chi-Square *X*^2^ Value	*p* Value
**Baseline Appointment Questions**				
*How difficult was it to remember everything you ate yesterday?*			5.41	0.14
Not at all difficult	11(73)	12(40)		
A little difficult	4(27)	13(43)		
Moderately difficult	˗	3(10)		
Very difficult	˗	2(7)		
*How did you feel using the ASA24 online system?*			2.23	0.14
Very comfortable	11(73)	15(50)		
Comfortable	4(27)	15(50)		
Uncomfortable	˗	˗		
Very uncomfortable	˗	˗		
**14-Day Appointment Questions**				
*How difficult was it to remember everything you ate on previous days?*			3.83	0.28
Not at all difficult	6(40)	14(47)		
A little difficult	9(60)	11(37)		
Moderately difficult	˗	3(10)		
Very difficult	˗	2(7)		
*How easy was it to access the ASA24 online system?*			1.28	0.73
Not at all easy	˗	2(7)		
A little easy	1(7)	3(10)		
Moderately easy	2(13)	3(10)		
Very easy	12(80)	22(73)		
*How did you feel using the ASA24 online system?*			4.43	0.22
Very comfortable	8(53)	16(53)		
Comfortable	4(27)	11(37)		
Uncomfortable	˗	2(7)		
Very uncomfortable	3(20)	1(3)		
*What was the HARDEST part of completing the ASA24 online dietary recall?*			0.2	0.98
Remembering to complete the questionnaire	2(13)	5(17)		
Remembering the food you ate	9(60)	16(53)		
Using the website	1(7)	2(7)		
Other	3(20)	7(23)		
*What was the EASIEST part of completing the ASA24 online dietary recall?*			0.83	0.66
Remembering to complete the questionnaire	6(40)	8(27)		
Remembering the food you ate	2(13)	5(17)		
Using the website	7(47)	17(57)		
*How helpful did you find the e-mail/text reminders?*			0.53	0.77
Not at all helpful	˗	˗		
A little helpful	˗	˗		
Moderately helpful	˗	1(3)		
Very helpful	6(40)	11(37)		
Extremely helpful	9(60)	18(60)		
*How did you complete MOST of your ASA24 dietary recalls?*			0.87	0.65
Smartphone	9(60)	16(53)		
Tablet	1(7)	5(17)		
Desktop computer	5(33)	9(30)		
*Did you need someone to help you complete the dietary recalls?*			1.61	0.21
Yes	˗	3(10)		
No	15(100)	27(90)		
*Which method do you think is best to complete a 24 h dietary recall?*			2.81	0.25
In-person with an interviewer	˗	2(7)		
Over the phone with an interviewer	˗	3(10)		
Online using ASA24	15(100)	25(83)		

## Data Availability

The data presented in this study are available on request from the corresponding author. The data are not publicly available due to institutional guidelines.

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
