# Peer review of "Feasibility, Acceptability, and Preliminary Validity of Self-Report Dietary Assessment in Adults with Multiple Sclerosis: Comparison with Doubly Labeled Water Measured Total Energy Expenditure"

_nutrients, 2021, doi:10.3390/nu13041198_

Round 1

Reviewer 1 Report

The study assessed the feasibility, acceptability and preliminary validity of self-report dietary assessment in adults with multiple sclerosis. The paper reads well and may be of interest to the readers, and especially to researchers who plan to use the ASA24 instrument in MS patients. There are several parts which may be improved, as described below.

In the Methods section, it is not clear how the sample size for this study was determined. Specifically, was some power analysis performed?

In the Results section, the first paragraph seems out of place: "This section may be divided by subheadings. It should provide a concise and precise description of the experimental results, their interpretation, as well as the experimental conclusions that can be drawn".

In the Results section, it is not clear how thr correlation analyses can be used to assess for potential bias?

In the Discussion section in paragraph on validity, discussion on correlation between the DLW TEE and ASA24 should include discussion on the sample size. This is related also to the power analysis mentioned above.

In funding, some wording seems out of place ("Please add", "by in part, by"). 

Author Response

Reviewer 1

The study assessed the feasibility, acceptability and preliminary validity of self-report dietary assessment in adults with multiple sclerosis. The paper reads well and may be of interest to the readers, and especially to researchers who plan to use the ASA24 instrument in MS patients. There are several parts which may be improved, as described below.

Thank you for your time and efforts reviewing our manuscript. We appreciate your suggestions for improving the overall quality and clarity.

In the Methods section, it is not clear how the sample size for this study was determined. Specifically, was some power analysis performed?

We updated the statement regarding the a-priori recruitment goal (L88-91). We clarify that the target recruitment numbers were aligned with previous research rather than a power analysis, and further based on the constraints of the pilot funding mechanism.

In the Results section, the first paragraph seems out of place: "This section may be divided by subheadings. It should provide a concise and precise description of the experimental results, their interpretation, as well as the experimental conclusions that can be drawn".

This portion was from the original template provided by MDPI and is now removed.

In the Results section, it is not clear how thr correlation analyses can be used to assess for potential bias?

The Pearson correlation analyses assessed the direction and magnitude of association between the measures as an indicator of preliminary validity. Assessment of potential bias was determined using Bland-Altman plots.

In the Discussion section in paragraph on validity, discussion on correlation between the DLW TEE and ASA24 should include discussion on the sample size. This is related also to the power analysis mentioned above.

This is a great point! We have now elaborated further in the Discussion regarding the sample size as a limitation and that the lack of a power analysis may have resulted in non-significant correlation coefficients (L319-322).

In funding, some wording seems out of place ("Please add", "by in part, by"). 

Thank you for this suggestion. We updated the wording per your suggestion (L378).

Reviewer 2 Report

What do you want to do ? New mailCopy

What do you want to do ? New mailCopy

Thank you for giving me the opportunity to review this article.

Do the conclusions really match the results with regards to validity, if we consider the poor correlation of the assessment of energy intake using ASA24 with TEE? This point would warrant further discussion.

How can we make sure intakes cover patients’ needs? It would seem useful to have some kind of assessment of undernutrition in patients, their level of physical activity, and whether patients’ had recently lost weight

Line 179: Please delete this paragraph from the original template: “This section may be divided by subheadings. It should provide a concise and precise 180 description of the experimental results, their interpretation, as well as the experimental 181 conclusions that can be drawn.”

Conclusions Line 335: “The current study ascertained the feasibility, acceptability, and validity of DLW and ASA24 diet recalls in persons with MS and provides a foundation regarding appropriate outcome tools for supporting dietary interventions.” This phrase doesn’t entirely match the aim of the study presented in introduction, which concerned validity and acceptability of the ASA24 questionnaire, and not DLW.  

Author Response

Reviewer 2

Thank you for giving me the opportunity to review this article.

Thank you for your thoughtful review of our manuscript. The manuscript is now revised, and we believe your suggestions, incorporated throughout, improved the overall quality.

Do the conclusions really match the results with regards to validity, if we consider the poor correlation of the assessment of energy intake using ASA24 with TEE? This point would warrant further discussion.

Per Reviewer #1’s suggestion, we elaborated further in the Discussion regarding the lack of a power analysis and the need for studies with larger samples when examining the validity of ASA24 (L319-322).

How can we make sure intakes cover patients’ needs? It would seem useful to have some kind of assessment of undernutrition in patients, their level of physical activity, and whether patients’ had recently lost weight

We think this is a great point and area for further research. This study ensured that the ASA24 was a feasible and acceptable tool for assessment of nutritional status in the future. We now address this point more directly in the Discussion (L358-361).

Line 179: Please delete this paragraph from the original template: “This section may be divided by subheadings. It should provide a concise and precise 180 description of the experimental results, their interpretation, as well as the experimental 181 conclusions that can be drawn.”

Thank you; this portion is now removed.

Conclusions Line 335: “The current study ascertained the feasibility, acceptability, and validity of DLW and ASA24 diet recalls in persons with MS and provides a foundation regarding appropriate outcome tools for supporting dietary interventions.” This phrase doesn’t entirely match the aim of the study presented in introduction, which concerned validity and acceptability of the ASA24 questionnaire, and not DLW.  

Thank you for highlighting this discrepancy. We updated the statement in the conclusion, and it now reflects the true aim of the study, “The current study ascertained the feasibility, acceptability, and validity of ASA24 diet recalls in persons with MS.” (L365)

Reviewer 3 Report

[General Comment]

The purpose of this study is to examine the feasibility acceptability, and validity of 24-hour dietary recall using online tool in adults with multiple sclerosis. This point is very important and is useful information in clinical setting, but there are major limitations to the analysis and unclear point of methods and results. In particular, please refer to the latest literature on the DLW method, and reconsider the method for calculating total energy expenditure for human.

[Major comments]

Methods

1) Line 112. With the current sample analysis precision with isotope ratio mass spectrometry, subjects are dosed with at least 1.8 g water/kg body water of water with 10% 18O atoms and 0.12 g water/kg body water of water with 99% 2H atoms (International Atomic Energy Agency 2009). Please describe the details of the processes of calculation, and evidences.

2) How did you calculated the daily energy intake from ASA24? Did you simply show the average value? Or did you consider the weekday and weekend?

3) Line 166. Sample size and power-analysis is lacking. How do you justify the number of participants of two groups?

4) What is the structure of the acceptability questionnaire? Is it a selection formula? Or free description? If the questionnaire include a selection formula, please describe it or show the questionnaire as figure.

Results

5) Line 179–181. This is a template of MDPI article. Please remove these sentences.

6) Line 196–198. You provided an additional 50 ml of water to take the urine sample at 4-hour. Did you subtract that 50 ml of water from the dilution spaces? How did you calculate the dilution spaces and total energy expenditure? Please explain more detail in the methods section.

7) Please justify why you excluded the results for 3 participants with MS.

8) Line 207–208. The total number of participants was 15 in HC and 30 in MS. The percentage of completing 4-5 ASA24 recalls in HC was 33%, and in MS was 20%. Although you described that the trends were similar between groups, please explain the reason it.

9) Table 1 and Table 2. Please show the statistical results. And align the line in table 2.

10) Figure 1 and Figure 2. Please show the horizontal axis scale.

11) How did you ensure that MS participants void of the bladder when urine sampling at baseline. When using urine samples with the DLW method, complete urination is a prerequisite.

Author Response

Reviewer 3

The purpose of this study is to examine the feasibility acceptability, and validity of 24-hour dietary recall using online tool in adults with multiple sclerosis. This point is very important and is useful information in clinical setting, but there are major limitations to the analysis and unclear point of methods and results. In particular, please refer to the latest literature on the DLW method, and reconsider the method for calculating total energy expenditure for human.

Thank you for your comprehensive and overall positive review of our manuscript. We appreciate your time reviewing the manuscript and suggestions for improving the reporting of the Methods and Results.

[Major comments]

Methods

1) Line 112. With the current sample analysis precision with isotope ratio mass spectrometry, subjects are dosed with at least 1.8 g water/kg body water of water with 10% 18O atoms and 0.12 g water/kg body water of water with 99% 2H atoms (International Atomic Energy Agency 2009). Please describe the details of the processes of calculation, and evidences.

We have now updated the Methods to clarify our calculation and the evidence (L116-120). “The solutions used were Cortecnet Oxygen-18 (H218O) Isotope Enrichment>=10% and Deuterium (D2O) 99.8% atom D. Further the dose administered was calculated: 18O:D2O is 1g:0.08g by weight, wherein participants were dosed 1g total solution per kg= 0.926g/kg of water with 10% 18O atoms and 0.074g/kg of water with 99.8% 2H atoms.”

2) How did you calculated the daily energy intake from ASA24? Did you simply show the average value? Or did you consider the weekday and weekend?

Thank you for highlighting this area of the Methods and providing feedback for improving. We now include further detail regarding the calculation of energy intake from ASA24, which included all valid days, irrespective of weekend or weekday (L156-157). Additionally, we added the need to further examine specifics such as weekend versus weekday intake into the Discussion of limitations as an area for further research in larger samples (L359-360).

3) Line 166. Sample size and power-analysis is lacking. How do you justify the number of participants of two groups?

The current study was a small pilot study aiming to provide a foundation and guide further research. Reviewer 1 provided similar feedback, which led us to update the statement regarding the a-priori recruitment goal (L88-91). We clarified that the target recruitment numbers were aligned with previous research, rather than a power analysis, and limited by the pilot funding mechanisms, and we recognize the small sample size as a limitation (L319-322).

4) What is the structure of the acceptability questionnaire? Is it a selection formula? Or free description? If the questionnaire include a selection formula, please describe it or show the questionnaire as figure.

The acceptability questionnaires included multiple-choice and open-ended responses. In Table 2, all multiple-choice questions and response options are provided in the first column and all questions are provided in the Methods. We updated the Methods to specify which questions were multiple choice and which were open ended in a response to clarify the questionnaire format (L163-177).

Results

5) Line 179–181. This is a template of MDPI article. Please remove these sentences.

Thank you; this portion is now removed.

6) Line 196–198. You provided an additional 50 ml of water to take the urine sample at 4-hour. Did you subtract that 50 ml of water from the dilution spaces? How did you calculate the dilution spaces and total energy expenditure? Please explain more detail in the methods section.

Thank you for this comment. Further discussion regarding the sample calculations led to several updates in the Methods and Results: (i). We now updated appropriate portions in the Results to clarify that 50ml was given at baseline for the first 20 participants and 125ml at baseline for the subsequent 25 participants to assist with dosed urine sample collection (L214-217). We recalculated TEE subtracting the appropriate quantity of water from each participant’s dilution space and the difference was negligible (i.e., mean=0.28%), therefore we did not change the values throughout. (ii). Calculations were based on the 3-hour sample for all participants, which is now clarified in the Results (L219-220). (iii). Equations for dilution space and TEE are now included in text (L134-139).

7) Please justify why you excluded the results for 3 participants with MS.

We now provided a statement clarifying that the decision to exclude the samples was based on the research team’s expertise regarding reasonable values for DLW TEE (L222-223).

8) Line 207–208. The total number of participants was 15 in HC and 30 in MS. The percentage of completing 4-5 ASA24 recalls in HC was 33%, and in MS was 20%. Although you described that the trends were similar between groups, please explain the reason it.

This is a valuable observation, however, we did not ascertain this information. Our interactions with this sample and 2 decades working with participants with MS have indicated that persons with MS are altruistically motivated for research on MS disease. This may have led to better compliance compared to healthy controls.

9) Table 1 and Table 2. Please show the statistical results. And align the line in table 2.

This is a great suggestion; we have now included p values in Table 1. Table 2 includes only descriptive results of questionnaire responses and statistical tests were not applicable. We have aligned text in Table 2.

10) Figure 1 and Figure 2. Please show the horizontal axis scale.

We added the horizontal axis scale back to the bottom of Figures 1 & 2.

11) How did you ensure that MS participants void of the bladder when urine sampling at baseline. When using urine samples with the DLW method, complete urination is a prerequisite.

We updated the Methods with a statement regarding the standard protocol used for voiding the bladder of urine. Indeed, participants were instructed to void the bladder to the best of their ability during baseline urine collection (L120-122).

Reviewer 4 Report

The authors present an article regarding feasibility, acceptability, and preliminary validity of self-report dietary assessment in multiple sclerosis patients. The examination of these indicators is very important for the scientific soundness of this questionnaire and its application to patients with neurological disorders. Overall, the work is appropriate, and the methodology appears robust for testing the feasibility, acceptability, and preliminary validity of this self-report.

Author Response

Reviewer 4

The authors present an article regarding feasibility, acceptability, and preliminary validity of self-report dietary assessment in multiple sclerosis patients. The examination of these indicators is very important for the scientific soundness of this questionnaire and its application to patients with neurological disorders. Overall, the work is appropriate, and the methodology appears robust for testing the feasibility, acceptability, and preliminary validity of this self-report.

Thank you for your time reviewing our manuscript and positive evaluation regarding the overall study aims and presentation.

Round 2

Reviewer 3 Report

[General Comment]

Thank you for your consideration correction of methods and results. Because you have used DLW method as reference of total energy expenditure to assess the validity of ASA 24, methodology of DLW is very important. However, I still found some issues in DLW methods. You need to explain about some limitation of methodology and consider the conclusion even this is a pilot study.

[Major comments]

Methods

1) Please correct the chemical formula. Mass is written as superscript. Atomic number is expressed as subscript. Example: D2O and H218O.

2) You should refer to the human studies. Speakman’s paper (Reference no. 18) described about basal metabolic rate for animals.

3) How did you calculate the CO2 production? You should recalculate using the deuterium dilution space by 4.1% and the 18O dilution space by 0.7% compared with body water according to latest reference.

4) How did you decide the DLW dose? Please justify that dose in this study was sufficient to measure total energy expenditure for 14 days.

Results

5) How did the research team’s expertise make a decision of excluding 3 participants? Are there some criteria such as inter-individual SD?

6) Please show the result of chi-square analysis in the table 2 or the manuscript.

Discussions and Conclusions

7) Do you think that saliva or blood in the DLW method is more appropriate for MS patients? If you have some opinion, please describe in the discussion section.

8) Validity of ASA 24 diet recalls is not clear in this study due to methodological issues. Please reconsider the discussions and conclusion.

Author Response

Reviewer 3

Thank you for your time and consideration in reviewing our resubmitted manuscript. We appreciate the focal feedback regarding the Methods, Results, and Discussion/Conclusion. We have addressed the major comments, as described below in bold, and using tracked changes within the manuscript.

[Major comments]

Methods

1) Please correct the chemical formula. Mass is written as superscript. Atomic number is expressed as subscript. Example: D2O and H218O.

Thank you for this observation, and we have now updated the chemical formulas (L117-121).

2) You should refer to the human studies. Speakman’s paper (Reference no. 18) described about basal metabolic rate for animals.

Our reference for Equation R2 of Speakman is now updated with the appropriate human study reference published in 2021, aligned with our interpretation Comment #3 below. The updated reference is: Speakman, J.R.; Yamada, Y.; Sagayama, H. A standard calculation methodology for human doubly labeled water studies. Cell Rep Med. 2021, 2, 100203.

3) How did you calculate the CO2 production? You should recalculate using the deuterium dilution space by 4.1% and the 18O dilution space by 0.7% compared with body water according to latest reference.

We have now updated the CO2 and TEE calculations to align with the updated Speakman et al. reference [18] (L134):

Equation 3: rCO2 = 0.4554 * N [1.007 * ko)] * 22.26

Equation 5: TEE (MJ/d) = rCO2 * (1.106 + (3.94/RQ) * (4.184/103)

4) How did you decide the DLW dose? Please justify that dose in this study was sufficient to measure total energy expenditure for 14 days.

The DLW dose was based on previous research within our laboratory and other groups and now include a reference: Hall, K. D.; Guo, J.; Chen, K. Y.; et al. Methodologic considerations for measuring energy expenditure differences between diets varying in carbohydrate using the doubly labeled water method. Am J Clin Nutr. 2019, 109, 1328-1334 (L121). Further, we inspected the baseline deuterium and 18O values compared to 14-day values to ensure there was not significant wash-out.

Results

5) How did the research team’s expertise make a decision of excluding 3 participants? Are there some criteria such as inter-individual SD?

The use of inter-individual SD is a great suggestion. We constructed box and whiskers plots and two of the values were identified as outliers with DLW TEE values greater than 2SDs above the sample mean. We updated the Methods and Results to reflect our approach for identifying outliers and final determination (L194-195;227-228).

6) Please show the result of chi-square analysis in the table 2 or the manuscript.

Table 2 now includes Chi-square analyses regarding acceptability questions between HC and MS participants. Additionally, we updated the Data analyses section to reflect the additional analyses (L188-189).

Discussions and Conclusions

7) Do you think that saliva or blood in the DLW method is more appropriate for MS patients? If you have some opinion, please describe in the discussion section.

This is a great suggestion for future research, thank you! We updated the Discussion with a recommendation for future research examining saliva or blood for assessing DLW in persons with MS considering bowel/bladder dysfunction (L351-353).

8) Validity of ASA 24 diet recalls is not clear in this study due to methodological issues. Please reconsider the discussions and conclusion.

We have now updated the Discussion and Conclusions and these sections more clearly reflect the preliminary nature of the study and importance of future research (L325-327;376-380).